# Analgesic Effects of Continuous Wound Infusion Combined with Intravenous Patient-Controlled Analgesia for Thoracic Surgery: A Retrospective Study

**DOI:** 10.3390/ijerph19116920

**Published:** 2022-06-06

**Authors:** Bo Hyun Jang, Keum Young So, Sang Hun Kim

**Affiliations:** 1Department of Medicine, Graduate School, Chosun University, 309 Pilmun-daero, Dong-gu, Gwangju 61452, Korea; sirius0503@naver.com; 2Department of Anesthesiology and Pain Medicine, Chosun University Hospital, 365 Pilmun-daero, Dong-gu, Gwangju 61453, Korea; kyso@chosun.ac.kr; 3Department of Anesthesiology and Pain Medicine, School of Medicine, Chosun University, 309 Pilmun-daero, Dong-gu, Gwangju 61452, Korea

**Keywords:** continuous wound infusion analgesia, local anesthetics, opioid analgesics, patient-controlled analgesia, postoperative pain, propensity score matching, thoracic surgery

## Abstract

Continuous wound infusion analgesia (CWA) with local anesthetics is a loco-regional anesthetic approach for multimodal analgesia management in surgical procedures. This study analyzed whether the combination of intravenous patient-controlled analgesia (PCA) and CWA would be more effective than PCA alone for postoperative analgesia and in preventing chronic postsurgical pain syndrome (PSPS) after thoracic surgeries. We enrolled 166 patients after propensity score matching, the PCA alone (PCA group, *n* = 83) and the combination of PCA and CWA (PCA-CWA group, *n* = 83), through a review of electronic medical records. The primary endpoint was the numeric rating scale (NRS) at postoperative days 1, 2, 3, 4, and 5. The secondary endpoint was the presence of PSPS at 3 and 6 months postoperatively. The NRS were lower in the PCA-CWA group than in the PCA group throughout the postoperative period (*p* < 0.001). The sedation incidence was lower in the PCA-CWA group (1.2%) than in the PCA group (9.6%) (*p* = 0.034), and there was no significant difference in other postoperative complications or in the incidence of PSPS (*p* = 1.000). The combination of intravenous PCA and CWA is an effective postoperative analgesic modality for thoracic surgery.

## 1. Introduction

Following thoracic surgery, patients frequently complain of postoperative pain, which induces poor respiratory effort and impaired pulmonary function, resulting in atelectasis, airway obstruction, shunting, and hypoxemia [1]. These postoperative complications result in a longer hospital stay due to the need for additional treatments. Furthermore, adequate postoperative analgesia is crucial for the prevention of chronic post-surgical pain syndrome (PSPS), because the incidence of PSPS is as high as 80% after 3 months and 75% after 6 months [1]. Therefore, effective postoperative analgesia with an ideal analgesic technique is paramount to prevent postoperative complications and promote early mobilization.

Various modalities of postoperative analgesia after thoracic surgeries have been attempted; however, there is no internationally accepted regimen that is considered the best strategy. The easiest and most common method is intravenous patient-controlled analgesia (PCA) using opioids [1]. However, it is difficult to achieve a balance between effective analgesia and undesirable effects such as respiratory depression, nausea, vomiting, ileus, and urinary retention [1,2]. Therefore, thoracic epidural analgesia has been considered the gold standard analgesic modality with superior analgesia, reduced opioid requirement, and greater patient satisfaction [1,3]. However, it is associated with several risk factors, including dural puncture, spinal cord damage, epidural hematoma, infection and abscess, hypotension, and urinary retention [1,3]. Inter-pleural and extra-pleural analgesia (paravertebral and intercostal block) have been reported to be valid alternatives to epidural analgesia [1,3]. However, there is a high risk of systemic toxicity with the use of local anesthetics, although these modalities are easier to administer and do not present the risks of opioid-related complications observed following systemic opioid and epidural analgesia [4].

Continuous wound infusion of local anesthetics (continuous wound infusion analgesia, CWA) through a multi-perforated catheter is a loco-regional anesthetic modality for multimodal analgesia management. CWA has been used to control postoperative pain as a safe and effective alternative modality in various surgeries with less pain and requirement for rescue opioids [5,6,7,8,9,10,11]. However, the analgesic effect of CWA is still unclear in patients undergoing thoracic surgeries, although several studies have reported postoperative analgesic effects of CWA alone or in combination with other postoperative analgesic modalities [2,12,13,14,15].

This study analyzed whether the combination of intravenous PCA and CWA would show more effective postoperative analgesia and fewer postoperative complications than PCA alone in patients undergoing thoracic surgery.

## 2. Materials and Methods

### 2.1. Study Design and Ethical Statement

The Institutional Review Board (IRB) of Chosun University Hospital approved this retrospective study based on an electronic medical record review (CHOSUN 2020-12-048) on 17 December 2020. The IRB also waived the need to obtain written informed consent from patients because the patients’ identification information was anonymized before the analysis, and this study did not pose any more than minimal risk to subjects. This study was prospectively registered with the Clinical Research Information Service (CRIS: https://cris.nih.go.kr/, ref: KCT0006804) accessed on 2 December 2021 and was conducted in accordance with the Declaration of Helsinki of 1964 and its subsequent revisions.

### 2.2. Selection of the Study Population

A total of 1658 patients were enrolled who received postoperative intravenous PCA alone (PCA group) or a combined modality of intravenous PCA and CWA (PCA-CWA group) after thoracic surgery from a same surgeon, after manual review of their electronic medical record, between 1 January 2010 and 30 November 2020 (Figure 1). The study included patients between the ages of 20 and 75 years who underwent a thoracotomy and open reduction/internal fixation (ORIF), as CWA was only applied for postoperative analgesia in patients who underwent these operations. Of the patients receiving ORIF, we included those with multiple rib fracture requiring large wound incision as large as thoracotomy, and sharing similar postoperative pain intensity [16]. This study excluded patients who received intravenous PCA without fentanyl, or were classified as American Society of Anesthesiologists physical status (ASA-PS) IV and V.

### 2.3. Interventions

#### 2.3.1. PCA

Postoperative PCA was performed according to the standardized protocol at our institution: The anesthesiologist operated the PCA device at the end of the surgery, which was set with a background infusion rate of 2 mL/h, bolus volume of 2 mL, and a lockout interval of 15 min. A total PCA volume of 100 mL, consisting of normal saline, fentanyl, adjuvant analgesics (nefopam or ketorolac), and adjuvant antiemetic (ramosetron), was used. The dose of each drug and the type of PCA device was decided considering the expected postoperative pain and the patient’s conditions. Postoperative pain with a numeric rating scale (NRS: 0 = no pain, 10 = worst pain) > 4 was treated by actioning the “demand” button to administer a preset bolus volume. Within the lockout interval, physicians or nurses administered the rescue analgesics such as opioids, nonsteroidal anti-inflammatory drugs, or other analgesics. Postoperative nausea and vomiting (PONV) with NRS > 4 were controlled with intravenous metoclopramide 10 mg or ramosetron 0.3 mg. Rescue analgesics and antiemetics were administered only on demand and not routinely. The severity of postoperative pain and PONV using NRS and any adverse events were recorded by nurses trained in the hospital to assess patients. Decisions to discontinue PCA were made by attending anesthesiologists based on the severity of the signs and symptoms of the patients.

#### 2.3.2. CWA

CWA was performed placing a multi-perforated wound catheter (Painfusor, Baxter, Maurepas, France) during wound closure, which the surgeon inserted from the lower end of the incision. The catheter was sutured as close as possible to the intercostal nerve and the deep surface of the serratus muscle throughout its entire length [13]. After administration of 10 mL of 0.25% ropivacaine at the end of the procedure, the catheter was connected to a continuously infusing container (Infusor LV, Baxter, Auckland, New Zealand), which allowed a 2.5 mg/mL of ropivacaine delivery at a constant flow rate of 2 mL/h for 5 days.

### 2.4. Outcomes

This study evaluated age, sex, weight, body mass index (BMI), ASA-PS, diabetes mellitus, hypertension, risk factors for PONV (smoking, motion sickness, and previous PONV), diagnosis, operation name, operation duration, duration of anesthesia, and length of hospital stay.

The PCA regimens (types and doses of opioids, adjuvant analgesics, and adjuvant antiemetics), type of PCA device, and the operating days of each analgesic modality were investigated. Doses of adjuvant analgesics were converted to fentanyl-equivalent doses (μg) considering the ratios of ketorolac (mg) to fentanyl (30:100) or nefopam (mg) to fentanyl (20:100) [17]. NRS was investigated, which were last recorded on postoperative days (POD) 0, 1, 2, 3, 4, and 5. The requirement for rescue analgesics and rescue antiemetics was investigated during the same period. Postoperative complications such as PONV, hypotension, dizziness, headache, pruritus, sedation, urinary retention, motor weakness, respiratory difficulty, and discontinuation of PCA were investigated 5 days after the operation. The presence of persistent pain was assessed at 3 and 6 months postoperatively.

### 2.5. Analysis

The primary endpoint was NRS at POD 1, 2, 3, 4, and 5. The secondary endpoint was the presence of persistent postoperative pain after 3 and 6 months.

All statistical analyses were performed using the SPSS Statistics for Windows, v 26.0 (IBM Corp., Armonk, NY, USA). All data are presented as means (95% confidence intervals [CI]) or as numbers (percentage) of patients (*n* patients (%)).

Patients who received intravenous PCA with CWA were matched to those who received intravenous PCA alone (control group) at a 1:1 ratio and 0.1 match tolerance using propensity score matching (PSM). This matching was used to obtain groups of patients corresponding to the two analgesic modalities that were balanced in terms of age, sex, body mass index, operation name, duration of anesthesia, dose of fentanyl used for intravenous PCA, types of adjuvant analgesics and NRS on POD 0.

Continuous variables were tested for normality using the Shapiro–Wilk test. Variables with non-skewed distributions were reported as means (95% CI), and differences were evaluated using the unpaired Student’s *t*-test. For the analysis of time interval data that passed Mauchly’s sphericity test, the author used repeated measures analysis of variance (ANOVA) and Wilk’s lambda multivariate ANOVA for data that did not pass Mauchly’s sphericity test. A one-way analysis of variance (ANOVA) test was used to compare the three groups at each time interval. Nominal variables were analyzed using the χ^2^ test or Fisher’s exact test. Statistical significance was established at *p* < 0.05.

## 3. Results

This study excluded 1177 of the 1658 enrolled in the PCA or PCA-CWA groups for the following reasons (Figure 1): patients who received surgical procedures other than thoracotomy and ORIF (1070 of the PCA group and 6 of the PCA-CWA group), patients under 20 years of age or over 75 years of age (357 in the PCA group and 43 in the PCA-CWA group), opioids other than fentanyl used for PCA (32 in the PCA group and 2 in the PCA-CWA group), and patients of ASA-PS IV and V classes (31 in the PCA group and 13 in the PCA-CWA group). Finally, this study enrolled a total of 481 patients, who complied with the aforementioned PCA and CWA protocol of this hospital, to compare postoperative analgesic effects between the PCA and PCA-CWA groups. A further 83 patients were selected from each group for analysis after PSM.

### 3.1. Demographic and Clinical Data

Significant differences in age (*p* = 0.008), ASA-PS class (*p* = 0.001), and surgical procedure (*p* < 0.001) were observed before PSM was performed (Table 1). After PSM, these differences were not statistically significant (Table 2). There were no patients with motion sickness or previous PONV in either group.

### 3.2. Postoperative Analgesia Modalities

There were significant differences in the doses of fentanyl used for PCA (*p* < 0.001), type of adjuvant analgesics (*p* < 0.001), and the adjuvant antiemetic doses (*p* < 0.001) before PSM (Table 3). Fentanyl and ramosetron were used for PCA in both groups. The fentanyl dose for PCA was higher in the PCA group than in the PCA-CWA group (*p* < 0.001) (Table 3). Nefopam was used more frequently as an adjuvant analgesic in the PCA-CWA group than in the PCA group (*p* < 0.001) (Table 3). Adjuvant analgesic doses converted to equivalent doses of fentanyl were higher in the PCA-CWA group than in the PCA group (*p* < 0.001) (Table 3). The ramosetron dose for the prevention of PONV during PCA was higher in the PCA-CWA group than in the PCA group (*p* < 0.001) (Table 3). After PSM, these differences were not significant, except for the dose of ramosetron (*p* < 0.001) (Table 4). The ramosetron dose for the prevention of PONV during PCA was higher in the PCA-CWA group than in the PCA group (*p* < 0.001) (Table 4).

### 3.3. Postoperative Analgesic Effect

NRS was lower in the PCA-CWA group than in the PCA group during the entire postoperative period, before PSM (*p* < 0.001) (Figure 2). After PSM, the NRS was lower in the PCA-CWA group than in the PCA group during the entire postoperative period, except for NRS on POD 0 (*p* < 0.001) (Figure 3).

### 3.4. Postoperative Outcomes

There were significant differences in the need for rescue analgesics, sedation, and urinary retention before the PSM was performed (Table 5). The incidence of rescue analgesic requirement and urinary retention was higher in the PCA-CWA group (55.7% and 28.7%, respectively) than in the PCA group (46.7% and 19.3%, respectively). The incidence of sedation was lower in the PCA-CWA group (0.8%) than in the PCA group (11.5%). After PSM, these differences were not significant, except for the incidence of sedation (*p* = 0.034) (Table 6). The incidence of sedation was lower in the PCA-CWA group (1.2%) than in the PCA group (9.6%). There were no significant differences in postsurgical pain syndrome, which was persistent pain at 3 and 6 months postoperatively (*p* = 1.000) (Table 5 and Table 6).

## 4. Discussion

This study showed that multimodal analgesia with PCA and CWA was more effective in reducing postoperative pain 1, 2, 3, 4, and 5 days postoperatively based on the data analysis after PSM. This study also showed a lower incidence of sedation in patients with multimodal analgesia with PCA and CWA compared with PCA alone. This study is meaningful in that it analyzed the postoperative analgesic and PSPS preventive effects of multimodal analgesia with PCA and CWA in patients undergoing thoracotomy and thoracic ORIF.

There are several studies on postoperative analgesic effects and postoperative complications of CWA in patients undergoing a thoracotomy [2,12,13,14,15,18]. CWA was performed by suturing the catheter as closely as possible to the intercostal nerve or fascia in patients who underwent standard muscle-sparing thoracotomy [2,13]. CWA with local anesthetic alone was as effective for postoperative analgesia with comparable analgesic effects as intravenous PCA and showed no significant differences in analgesic effects and rescue analgesic requirements [2]. Furthermore, the approach presented benefits including a reduction in drowsiness, dizziness, respiratory depression, and decreased ICU stay and hospital expenditure [2]. The postoperative analgesic effect of the combined intravenous PCA and CWA modality was also investigated in patients who underwent thoracotomy, compared to intravenous PCA alone or other locoregional anesthetic modalities [13,14,15]. This combined modality showed a greater reduction in postoperative pain and analgesic requirements and achieved a faster recovery of respiratory function than intravenous PCA alone [13,15]. However, this modality did not show a significant benefit in reducing postoperative pain scores or opioid consumption compared to other locoregional anesthetic modalities or those combined with intravenous PCA [14,15]. Ultimately, the results of this study support previous studies, which showed that the combined modality of intravenous PCA and CWA was more effective in significantly reducing postoperative pain than PCA alone, although it was not superior to thoracic epidural analgesia, a combination modality of continuous thoracic paravertebral block with intravenous PCA or intravenous PCA alone [2,12,13,14,15,18].

The risk of local anesthetic systemic toxicity is higher with continuous peripheral nerve blockade than with single-shot techniques due to local anesthetic accumulation [19]. Thus, an important issue to consider in the clinical application of CWA is the adverse effects related to local anesthesia because the CWA catheter is placed as close as possible to the intercostal nerve, and the local anesthetic is continuously infused. Previous studies on the analgesic effect of CWA have described continuous infusion between 4 and 10 mg/h of ropivacaine or bupivacaine [2,13,14,15], with no toxicity related to local anesthetics [14,15,20]. The plasma concentration of bupivacaine increased continuously after 4 mg/h for 48 h postoperatively until the end of infusion, but the dose achieved was less than 4 μg/mL (toxic level) [13]. Other complications such as drowsiness and dizziness, sedation, cardiovascular effects, respiratory depression, infection, PSPS, and general complications were significantly lower or showed no differences in patients receiving CWA compared to other analgesic modalities [2,14,15]. This study did not show any local anesthetic-related adverse effects, although plasma ropivacaine concentrations were not measured and there were no complications in patients receiving 5 mg/h of ropivacaine.

This study has several limitations. First, there may be an inherent bias due to retrospective data collection through the review of electronic medical records. Thus, to minimize this bias, we analyzed the data with sufficient power after adjustment of demographic data, the type and dose of opioid used for postoperative analgesia, the type of surgical intervention, and the postoperative pain score on the day of surgery using PSM. However, the retrospective analysis might have influenced the results of this study, although the data were adjusted using PSM [14]. Second, patients who received CWA alone were not enrolled because there were no patients who received CWA alone. Therefore, it cannot be demonstrated whether CWA alone may be effective in controlling acute pain after thoracic surgery compared to other analgesic modalities, and whether CWA has a synergistic effect with other analgesic modalities [13]. Third, postoperative rescue analgesic requirements and opioid consumption were not investigated. Although the additional application of CWA to PCA showed a more significant postoperative pain reduction in PCA alone, this study revealed that both groups had received clinically effective postoperative analgesia. However, sedation, an opioid-related complication, was significantly higher in the PCA-only group than in the group that additionally used CWA for PCA. This can be explained by the possibility that more opioids were self-administered by using the PCA device with greater frequency in the PCA-only group due to postoperative pain [2].

## 5. Conclusions

This study demonstrates that the combined modality of intravenous PCA and CWA is a suitable option with effective and safe postoperative analgesia after thoracotomy and ORIF in patients with multiple rib fractures. However, this study, as supported by previous literature, suggests that it remains necessary to confirm whether the combined modality of intravenous PCA and CWA is more effective than intravenous PCA or other analgesic modalities, and whether it achieves a synergistic analgesic effect.

## Figures and Tables

**Figure 1 ijerph-19-06920-f001:**
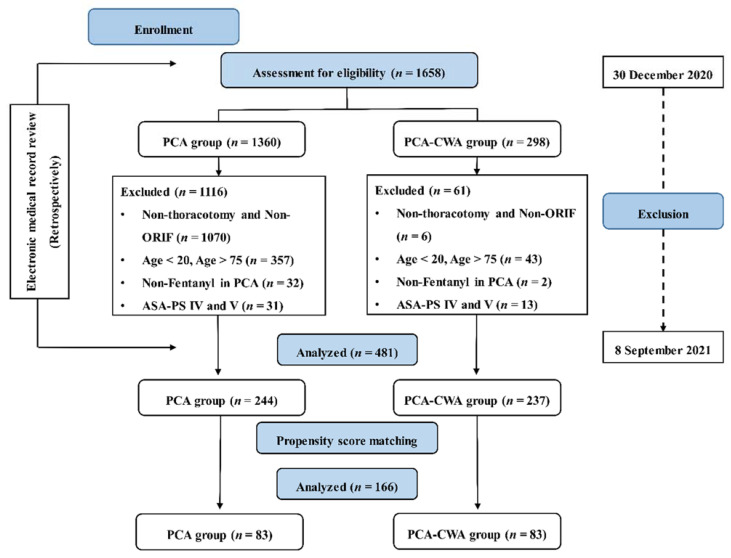
Flowchart of this study. ASA-PS, American Society of Anesthesiologists—Physical Status; ORIF, open reduction and internal fixation; PCA, intravenous patient-controlled analgesia. PCA group, group of patients who received postoperative intravenous PCA alone; PCA-CWA group, group of patients who received postoperative intravenous PCA and continuous wound infusion analgesia.

**Figure 2 ijerph-19-06920-f002:**
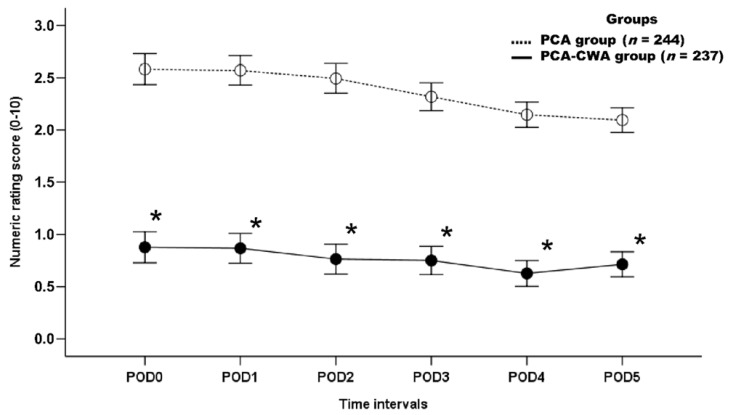
Numeric rating score over the 5-day postoperative course, before propensity score matching. POD, postoperative day; POD0, day of surgery. PCA group, group of patients who received postoperative intravenous PCA alone; PCA-CWA group, group of patients who received postoperative intravenous PCA and continuous wound infusion analgesia. * *p* < 0.05 is considered statistically significant.

**Figure 3 ijerph-19-06920-f003:**
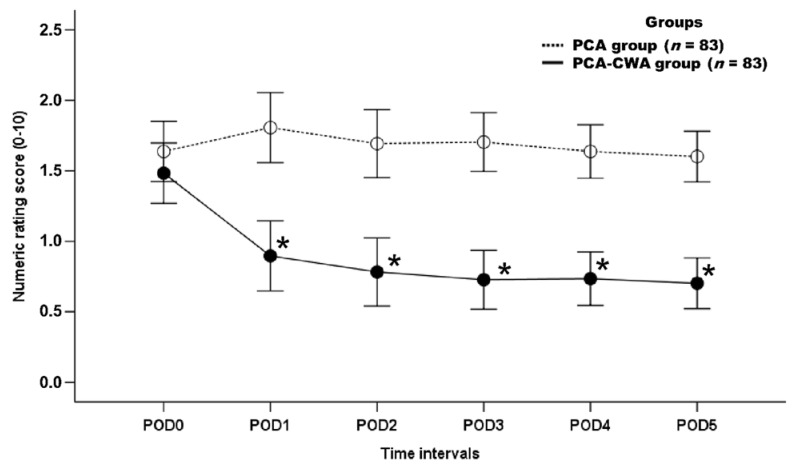
Numeric rating score at postoperative day 5, after propensity score matching. POD, postoperative day; POD0, day of surgery. PCA group, group of patients who received postoperative intravenous PCA alone; PCA-CWA group, group of patients who received postoperative intravenous PCA and continuous wound infusion analgesia. * *p* < 0.05 was considered statistically significant.

**Table 1 ijerph-19-06920-t001:** Demographic and clinical data before propensity score matching.

	PCA Group (*n* = 244)	PCA-CWA Group (*n* = 237)	*p*-Value
Age (year)	57.2 (55.7–58.8)	60.1 (58.7–61.5)	0.008 *
Sex (male/female)	188/56 (77/23)	180/57 (75.9/24.1)	0.776
Weight (kg)	64.1 (62.6–65.5)	63.9 (62.4–65.4)	0.866
Height (cm)	165.4 (164.3–166.6)	165.4 (164.3–166.5)	0.971
BMI (kg/m^2^)	23.3 (22.9–23.8)	23.3 (22.8–23.7)	0.868
ASA-PS (I/II/III)	49/142/53 (20.1/58.2/21.7)	25/133/79 (10.5/56.1/33.3)	0.001 *
Hypertension (yes)	67 (27.5)	84 (35.4)	0.059
Diabetic mellitus (yes)	45 (18.4)	54 (22.8)	0.239
Smoking (yes)	78 (32)	69 (29.1)	0.497
Duration of anesthesia (min)	174.6 (165.2–184.1)	183 (174.2–191.8)	0.202
Duration of operation (min)	149.5 (140.3–158.6)	157.7 (149–166.5)	0.198
Hospital stay (d)	19.5 (17.9–21.1)	21.7 (18.9–24.5)	0.172
Surgical procedures (ORIF/thoracotomy)	119/125 (48.8/51.2)	64/173 (27/73)	<0.001 *

Values are expressed as mean (95% confidence interval) or number (percentage) of patients. ASA-PS, American Society of Anesthesiologists-physical status; BMI, body mass index; CWA, continuous wound infusion analgesia; ORIF, open reduction and internal fixation; PCA: patient-controlled analgesia. PCA group, group of patients who received postoperative intravenous PCA alone; PCA-CWA group, group of patients who received postoperative intravenous PCA and continuous wound infusion analgesia. * *p* < 0.05 was considered statistically significant.

**Table 2 ijerph-19-06920-t002:** Demographic and clinical data after propensity score matching.

	PCA Group (*n* = 83)	PCA-CWA Group (*n* = 83)	*p*-Value
Age (year)	56.9 (54.1–59.7)	58.1 (55.7–60.6)	0.514
Sex (male/female)	59/24 (71.1/28.9)	59/24 (71.1/28.9)	1.000
Weight (kg)	63 (60.2–65.8)	64.6 (62–67.3)	0.406
Height (cm)	164.3 (161.8–166.7)	166.1 (164.2–167.9)	0.258
BMI (kg/m^2^)	23.2 (22.4–24.1)	23.3 (22.6–24.1)	0.804
ASA-PS (I/II/III)	8/53/22 (9.6/63.9/26.5)	14/43/26 (16.9/51.8/31.3)	0.222
Hypertension (yes)	26 (31.3)	27 (32.5)	0.868
Diabetic mellitus (yes)	16 (19.3)	16 (19.3)	1.000
Smoking (yes)	26 (31.3)	22 (26.5)	0.493
Duration of anesthesia (min)	181.5 (163.8–199.3)	183.2 (170.1–196.3)	0.879
Duration of operation (min)	160.3 (142.8–177.9)	156.6 (143.5–169.6)	0.734
Hospital stay (d)	21.8 (18.1–25.5)	20.3 (17–23.6)	0.547
Surgical Procedures (ORIF/thoracotomy)	44/39 (53/47)	37/46 (44.6/55.4)	0.277

Values are expressed as mean (95% confidence interval) or number (percentage) of patients. ASA-PS, American Society of Anesthesiologists-physical status; BMI, body mass index; CWA, continuous wound infusion analgesia; ORIF, open reduction and internal fixation; PCA: patient-controlled analgesia. PCA group, group of patients who received postoperative intravenous PCA alone; PCA-CWA group, group of patients who received postoperative intravenous PCA and continuous wound infusion analgesia.

**Table 3 ijerph-19-06920-t003:** Postoperative analgesia modalities before propensity score matching.

	PCA Group (*n* = 244)	PCA-CWA Group (*n* = 237)	*p*-Value
Fentanyl used for PCA	244 (100)	237 (100)	1.000
Doses (μg)	1107 (1075.2–1138.8)	944.7 (922–967.5)	<0.001 *
Adjuvant analgesics (nefopam/ketorolac/none)	107/93/44 (43.9/38.1/18)	228/4/5 (96.2/1.7/2.1)	<0.001 *
Dose (μg)	526.6 (490.5–562.8)	705.1 (685.3–724.8)	<0.001 *
Adjuvant antiemetics (ramosetron)	244 (100)	237 (100)	1.000
Dose (mg)	0.9 (0.9–0.9)	1.2 (1.2–1.2)	<0.001 *

Values are expressed as mean (95% confidence interval) or number (percentage) of patients. CWA, continuous wound infusion analgesia; PCA, patient-controlled analgesia. PCA group, group of patients who received postoperative intravenous PCA alone; PCA-CWA group, group of patients who received postoperative intravenous PCA and continuous wound infusion analgesia. * *p* < 0.05 was considered statistically significant.

**Table 4 ijerph-19-06920-t004:** Postoperative analgesia modalities after propensity score matching.

	PCA Group (*n* = 83)	PCA-CWA Group (*n* = 83)	*p*-Value
Fentanyl used for PCA	83 (100)	83 (100)	1.000
Dose (μg)	986.7 (935.2–1038.3)	974.7 (936.7–1012.6)	0.709
Adjuvant analgesics (nefopam/ketorolac/none)	71/2/10 (85.5/2.4/12)	74/4/5 (89.2/4.8/6)	0.302
Dose (μg) ^†^	643.4 (584.4–702.4)	691.6 (22.3–647.2)	0.196
Adjuvant antiemetics (ramosetron)	83 (100)	83 (100)	1.000
Dose (mg)	1 (1–1.1)	1.2 (1.2–1.2)	<0.001 *

Values are expressed as mean (95% confidence interval) or number (percentage) of patients. CWA, continuous wound infusion analgesia; PCA, patient-controlled analgesia. PCA group, group of patients who received postoperative intravenous PCA alone; PCA-CWA group, group of patients who received postoperative intravenous PCA and continuous wound infusion analgesia. * *p* < 0.05 was considered statistically significant. ^†^ Doses of fentanyl equivalents (μg) converted from doses of adjuvant analgesics with ratios of ketorolac (mg) to fentanyl (30:100), ratio of nefopam (mg) to fentanyl (1:5) [17].

**Table 5 ijerph-19-06920-t005:** Postoperative outcomes before propensity score matching.

	PCA Group (*n* = 244)	PCA-CWA Group (*n* = 237)	*p*-Value
PONV (yes)	15 (6.1)	10 (4.2)	0.341
Rescue analgesics (yes)	114 (46.7)	132 (55.7)	0.049 *
Rescue antiemetics (yes)	10 (4.1)	12 (5.1)	0.613
Hypotension (yes)	12 (4.9)	4 (1.7)	0.072
Dizziness (yes)	9 (3.7)	4 (1.7)	0.261
Headache (yes)	2 (0.8)	0 (0)	0.499
Pruritus (yes)	3 (1.2)	2 (0.8)	1.000
Sedation (yes)	28 (11.5)	2 (0.8)	<0.001 *
Urinary retention (yes)	47 (19.3)	68 (28.7)	0.015 *
Motor weakness (yes)	0 (0)	1 (0.4)	0.493
Respiratory difficulty (yes)	7 (2.9)	5 (2.1)	0.772
PCA stop (yes)	29 (3.7)	9 (3.8)	0.950
Persistent pain after 3 months postoperatively (yes)	0 (0)	2 (0.8)	0.242
Persistent pain after 6 months postoperatively (yes)	0 (0)	2 (0.8)	0.242

Values are expressed as number (percentage) of patients. CWA, continuous wound infusion analgesia; PCA: patient-controlled analgesia; PONV, postoperative nausea and vomiting. PCA group, group of patients who received postoperative intravenous PCA alone; PCA-CWA group, group of patients who received postoperative intravenous PCA and continuous wound infusion analgesia. * *p* < 0.05 was considered statistically significant.

**Table 6 ijerph-19-06920-t006:** Postoperative outcomes after propensity score matching.

	PCA Group (*n* = 83)	PCA-CWA Group (*n* = 83)	*p*-Value
PONV (yes)	4 (4.8)	2 (1.2)	0.682
Rescue analgesics (yes)	36 (43.4)	38 (45.8)	0.755
Rescue antiemetics (yes)	2 (2.4)	4 (4.8)	0.682
Hypotension (yes)	2 (2.4)	1 (1.2)	1.000
Dizziness (yes)	1 (1.2)	0 (0)	1.000
Headache (yes)	0 (0)	0 (0)	1.000
Pruritus (yes)	1 (1.2)	0 (0)	1.000
Sedation (yes)	8 (9.6)	1 (1.2)	0.034 *
Urinary retention (yes)	18 (21.7)	25 (30.1)	0.215
Motor weakness (yes)	0 (0)	0 (0)	1.000
Respiratory difficulty (yes)	6 (7.2)	1 (1.2)	0.117
PCA stop (yes)	3 (3.6)	5 (6)	0.720
Persistent pain after 3 months postoperatively (yes)	0 (0)	1 (1.2)	1.000
Persistent pain after 6 months postoperatively (yes)	0 (0)	1 (1.2)	1.000

Values are expressed as number (percentage) of patients. CWA, continuous wound infusion analgesia; PCA: patient-controlled analgesia; PONV, postoperative nausea and vomiting. PCA group, group of patients who received postoperative intravenous PCA alone; PCA-CWA group, group of patients who received postoperative intravenous PCA and continuous wound infusion analgesia. * *p* < 0.05 was considered statistically significant.

## Data Availability

The data presented in this study are available on request from the corresponding author, through the institutional review board, and reviewers. The data are not publicly available due to restrictions of obtaining approval from the institutional review board for the disclosure of data. If the data from this study are required, please do not hesitate to contact the corresponding author.

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
