# Peer review of "Analgesic Effects of Continuous Wound Infusion Combined with Intravenous Patient-Controlled Analgesia for Thoracic Surgery: A Retrospective Study"

_ijerph, 2022, doi:10.3390/ijerph19116920_

Round 1
Reviewer 1 Report
Dear Authors,
it is an interesting paper but in my opinion, it needs some corrections:
line 63-68 there is repetition - it needs re-editing
line 83 - add mean age
line 85. it is not clear to me. Does it mean that all analyzed patients (PCA & PAC +CWA) were treated due to multiple rib fractures and thoracotomy with ORIF were performed?
line 125 - smoking means more than 10? 20 cigarettes a day? Was I noted?
line 175 - "PCA group, postoperative intravenous PC" - make this explanation clear -like" group of patients who received postoperative intravenous PCA
alone" the same with "; PCA-CWA group, postoperative intravenous PCA and continuous wound infusion analgesia" - please apply in all tables
line 228 - in my opinion, the discussion needs re-editing and complementing. eg. "Ultimately, previous studies support the results of this study..." it seems to be opposite: that this study support previous ones.; or:
"This study is meaningful in that it analyzed (...) patients undergoing thoracotomy and thoracic ORIF." on the other hand authors write in the section with limitations: " this study included not only thoracotomy but also ORIF, as ORIF procedures ..." so is it "limitation" or meaningful? - it can be confusing
Author Response
Date of submission: 27th, May, 2022
Prof. Dr. Paul B. Tchounwou
Editor-in-Chief
International Journal of Environmental Research and Public Health
Dear Editor
Thank you for giving me the opportunity to submit a revised draft of my manuscript titled “Analgesic Effects of Continuous Wound Infusion Combined with Intravenous Patient-Controlled Analgesia for Thoracic Surgery: A Retrospective Study” to the International Journal of Environmental Research and Public Health. The manuscript ID is ijerph-1691830.
We appreciate the time and effort that you and the reviewers have dedicated to providing your valuable feedback on this manuscript. We are grateful to the reviewers for their insightful comments on this paper. We have been able to incorporate changes to reflect most of the suggestions provided by the reviewers. We have highlighted the changes within the manuscript using the "Track Changes" function in Microsoft Word. The responses to all the comments have been prepared and are given below.
Finally, if we get the final acceptance for this manuscript, we will upload the final manuscript after receiving the English editing from the native English speaker again.
Thank you for your consideration. I look forward to hearing from you.
Sincerely,
Sang Hun Kim, MD, PhD
Department of Anesthesiology and Pain Medicine, Chosun University, School of Medicine, 309 Pilmun-daero, Dong-gu, Gwangju 61453, Republic of Korea
+ 82-62-2203223
+ 82-62-2232333
ksh3223@chosun.ac.kr
Here is a point-by-point response to the reviewers’ comments and concerns.
Response to Reviewer 1 Comments
- We used the line number in the manuscript, which maintained the "Track Changes" function in Microsoft Word.
Point 1: line 63-68 there is repetition - it needs re-editing.
Response 1: Thank you for pointing this out. According to the reviewer's comment, we deleted and revised the last paragraph of the introduction (LL 63-65).
“This study hypothesized that the combined modality of intravenous PCA and CWA would show more effective postoperative analgesia and fewer postoperative complications than PCA alone in patients undergoing thoracic surgery. This study analyzed whether the combination of intravenous PCA and CWA would show more effective postoperative analgesia and fewer postoperative complications than PCA alone in patients undergoing thoracic surgery surgeries.”
Point 2: line 83 - add mean age.
Response 2: Thank you for pointing this out. This is a description of the enrolment of patients aged 20 to 75 who received thoracic surgeries for the purpose of this study. As commented by the reviewer, the authors also think that it is a good opinion to suggest the average age. But this is a description of the criteria for patient selection, and the authors don't think it's an important point to present the average age, so we decide not to label it. Please consider this point.
Point 3: line 85. It is not clear to me. Does it mean that all analyzed patients (PCA & PAC +CWA) were treated due to multiple rib fractures and thoracotomy with ORIF were performed?
Response 3: Thank you for pointing this out. The authors have revised this more clearly as follows in lines 83-85 of a revised manuscript.
“Of the patients receiving ORIF, we included those with multiple rib fracture requiring large wound incision as large as thoracotomy and shares similar postoperative pain intensity [16].”
Point 4: line 125 - smoking means more than 10? 20 cigarettes a day? Was I noted?
Response 4: Thank you for pointing this out. As the reviewer mentioned, how many cigarettes a day is an important part of defining 'smoking'. In general, this hospital defines and classifies smokers who continue to smoke more than 10 cigarettes a day. However, considering that most papers generally did not specifically describe the definition of smokers, the authors decided not to add additional definition of smokers. Please consider this point.
Point 5: line 175 - "PCA group, postoperative intravenous PC" - make this explanation clear -like" group of patients who received postoperative intravenous PCA alone" the same with "; PCA-CWA group, postoperative intravenous PCA and continuous wound infusion analgesia" - please apply in all tables.
Response 5: Thank you for pointing this out. The authors have revised this more clearly as follows in figures and tables of a revised manuscript.
“PCA group, group of patients who received postoperative intravenous PCA alone; PCA-CWA group, group of patients who received postoperative intravenous PCA and continuous wound infusion analgesia.”
Point 5: line 228 - in my opinion, the discussion needs re-editing and complementing. eg. "Ultimately, previous studies support the results of this study..." it seems to be opposite: that this study support previous ones.; or: "This study is meaningful in that it analyzed (...) patients undergoing thoracotomy and thoracic ORIF." on the other hand authors write in the section with limitations: " this study included not only thoracotomy but also ORIF, as ORIF procedures ..." so is it "limitation" or meaningful? - it can be confusing.
Response 5: Thank you for pointing this out. The authors have revised this more clearly as follows in a revised manuscript. Especially, for second question, the contents of the second limitation described in discussion have been deleted.
In lines 304-305 of a revised manuscript: “Ultimately, the results of this study support previous studies.”
In lines 337-341 of a revised manuscript: “Second, unlike other previous studies, this study included not only thoracotomy but al-so ORIF, as ORIF procedures for multiple rib fracture requires a large wound incision to the same extent as thoracotomy and shares similar postoperative pain intensity[20]. Third, patients who received CWA alone were not enrolled because there were no patients who received CWA alone. Therefore, it cannot be demonstrated whether CWA alone may be effective in controlling acute pain after thoracic surgery compared to other analgesic modalities, and whether CWA has a synergistic effect with other analgesic modalities [18]. Third, “
We look forward to hearing from you in due time regarding our submission and to responding to any further questions and comments you may have.
Sincerely,
[Sang Hun Kim, 27th, May, 2022]

Reviewer 2 Report
In this study Jang et al report a retrospective chart review comparing analgesia, complications and chronic pain after thoracic surgery with PCA or PCA + CWA. They report results comparing all patients identified meeting their inclusion criteria and then a second set of results which have been propensity score matched to balance based on demographic and operative features. They report decreased post-operative pain based on numeric rating scale (NRS) in the PCA+CWA group. They also report lower sedation levels in the PCA+CWA group and no other significant differences in post-operative complications acutely or in the development of chronic pain at 3 and 6 months. The study reports a valuable comparison from their retrospective review which may be useful in informing analgesic management in thoracic and other surgeries. However, some important information and details are lacking from the manuscript.
- In the introduction, it was unclear to me why the hypothesis that chronic pain syndromes (PSPS) would be reduced with adequate pain control post-operatively was made. It would be helpful to expand on the findings reported in reference 1 (early pain control reduces chronic pain) and to add additional references in your introduction to explain this phenomenon since it is reported as a secondary end point of the study.
- Additional details are needed on the initial identification of eligible patients for the study. What specific parameters were used to identify the 1658 patients enrolled? In addition were the outcomes automatically pulled from the electronic medical records or were the charts manually reviewed?
- The authors present CWA in conjunction with PCA as a new possible management for thoracic surgery; however, their study assessed patients who have already been receiving PCA+CWA between 2010-2020. I am curious to know more about why CWA is being used in this population of patients already and what factors may have influenced the managing team of physicians to select PCA+CWA vs PCA alone in these patients in the first place. This would be important to help understand whether there are any further biases in the patients that were selected for CWA that might impact the results. It would be helpful to understand more and for this to be addressed in some part of the manuscript (introduction or discussion).
- In the methods section reporting the PCA and CWA protocol, I am wondering if these exact parameters were used for all the patients included or if this is the standard protocol being reported? Presumably there may have been some exceptions to the standard protocol. It would be helpful to report whether all charts were reviewed and the protocol was followed exactly or whether this is the standard and there may have been some deviations.
- Additional details on the pain numerical rating scale are needed. Was this based on a standardized NRS (reference?) I assume it is self-reported by the patients and recorded in the charts. Are their any missing timepoints for patients? Who reported the NRS, a nurse, physician? How was this collected from the patients chart?
- Are the time points of 3 and 6 months for assessment of PSPS standard? Is it possible no differences were seen as it is too early to assess. The incidence in the cohort. Would be helpful to add to discussion.
- The discussion needs to be edited.
The discussion includes a lot of details that would be better suited in the introduction. Page 10 lines 236-250 should be in the introduction. The authors should use the discussion to further the implications of their results and discuss further as opposed to reporting background information.
The next two sections are better suited to discussion.
I am curious why CWA along was not included, were there too few? This would be an
interesting alternate management to discuss more in the discussion.
Additional sections in the discussion could be added based on the previous comments.
Author Response
Date of submission: 27th, May, 2022
Prof. Dr. Paul B. Tchounwou
Editor-in-Chief
International Journal of Environmental Research and Public Health
Dear Editor
Thank you for giving me the opportunity to submit a revised draft of my manuscript titled “Analgesic Effects of Continuous Wound Infusion Combined with Intravenous Patient-Controlled Analgesia for Thoracic Surgery: A Retrospective Study” to the International Journal of Environmental Research and Public Health. The manuscript ID is ijerph-1691830.
We appreciate the time and effort that you and the reviewers have dedicated to providing your valuable feedback on this manuscript. We are grateful to the reviewers for their insightful comments on this paper. We have been able to incorporate changes to reflect most of the suggestions provided by the reviewers. We have highlighted the changes within the manuscript using the "Track Changes" function in Microsoft Word. The responses to all the comments have been prepared and are given below.
Finally, if we get the final acceptance for this manuscript, we will upload the final manuscript after receiving the English editing from the native English speaker again.
Thank you for your consideration. I look forward to hearing from you.
Sincerely,
Sang Hun Kim, MD, PhD
Department of Anesthesiology and Pain Medicine, Chosun University, School of Medicine, 309 Pilmun-daero, Dong-gu, Gwangju 61453, Republic of Korea
+ 82-62-2203223
+ 82-62-2232333
ksh3223@chosun.ac.kr
Here is a point-by-point response to the reviewers’ comments and concerns.
Response to Reviewer 2 Comments
- We used the line number in the manuscript, which maintained the "Track Changes" function in Microsoft Word.
Point 1: In the introduction, it was unclear to me why the hypothesis that chronic pain syndromes (PSPS) would be reduced with adequate pain control post-operatively was made. It would be helpful to expand on the findings reported in reference 1 (early pain control reduces chronic pain) and to add additional references in your introduction to explain this phenomenon since it is reported as a secondary end point of the study.
Response 1: Thank you for pointing this out. We agree with this comment. The part specified by PSPS as a second end point of this study was revise as an integrated analysis of overall postoperative complications. So, we did not provide any additional references. Reflecting the comments of you and other reviewers, the authors revised the last paragraph of introduction with hypothesis as follows (in lines 63-65 of a revised manuscript). Please consider this point.
“This study hypothesized that the combined modality of intravenous PCA and CWA would show more effective postoperative analgesia and fewer postoperative complications than PCA alone in patients undergoing thoracic surgery. This study analyzed whether the combination of intravenous PCA and CWA would show more effective postoperative analgesia and fewer postoperative complications than PCA alone in patients undergoing thoracic surgery surgeries.”
Point 2: Additional details are needed on the initial identification of eligible patients for the study. What specific parameters were used to identify the 1658 patients enrolled? In addition were the outcomes automatically pulled from the electronic medical records or were the charts manually reviewed?
Response 2: Thank you for pointing this out. We agree with this comment. The authors extracted patients using each insurance claim code for intravenous PCA and CWA, and collected data by manually reviewing their electronic medical records. The authors described these information as follows in lines 79-80 of a revised manuscript.
“A total of 1658 patients were enrolled who received postoperative intravenous PCA alone (PCA group) or a combined modality of intravenous PCA and CWA (PCA-CWA group) after thoracic surgery from a same surgeon, under manually review of their electronic medical record, between 1 January 2010 and 30 November 2020 (Figure 1).”
Point 3: The authors present CWA in conjunction with PCA as a new possible management for thoracic surgery; however, their study assessed patients who have already been receiving PCA+CWA between 2010-2020. I am curious to know more about why CWA is being used in this population of patients already and what factors may have influenced the managing team of physicians to select PCA+CWA vs PCA alone in these patients in the first place. This would be important to help understand whether there are any further biases in the patients that were selected for CWA that might impact the results. It would be helpful to understand more and for this to be addressed in some part of the manuscript (introduction or discussion).
Response 3: Thank you for pointing this out. We agree with this comment. IV PCA has generally been used in this hospital as the primary modality of postoperative pain management. However, the usefulness of continuous wound infusion with local anesthetics for management of various postoperative pains has been suggested, and the hospital's cardiothoracic surgeons have also been willing to apply it. Therefore, since 2017, this hospital has been used after deciding whether to apply it to patients according to the judgment of a surgeon. As noted by the reviewer, it is agreed that such arbitrary application of CWA to certain patients without using a randomized controlled allocation method may result in some bias. In particular, in consideration of the possibility that the bias mentioned by the reviewer may occur in the retroactive study, the authors tried to minimize the bias by applying the PSM method. The authors believe that readers of this manuscript will understand this methodological content well. Therefore, it is judged that there is no need to additionally address these information in some part of the manuscript (introduction or discussion). Please consider this point.
Point 4: In the methods section reporting the PCA and CWA protocol, I am wondering if these exact parameters were used for all the patients included or if this is the standard protocol being reported? Presumably there may have been some exceptions to the standard protocol. It would be helpful to report whether all charts were reviewed and the protocol was followed exactly or whether this is the standard and there may have been some deviations.
Response 4: Thank you for pointing this out. We agree with this comment. We analyzed data from 481 patients, who complied with the hospital's PCA and CWA protocol mentioned in the 'intervention' of 'method', among 1658 patients. This is further described in lines 184-188 of this revised manuscript as follows. Please consider this point.
“Finally, this study enrolled a total of 481 patients, who complied with the aforementioned PCA and CWA protocol of this hospital, to compare postoperative analgesic effects between the PCA and PCA-CWA groups.”
Point 5: Additional details on the pain numerical rating scale are needed. Was this based on a standardized NRS (reference?) I assume it is self-reported by the patients and recorded in the charts. Are there any missing time points for patients? Who reported the NRS, a nurse, physician? How was this collected from the patients chart?
Response 5: Thank you for pointing this out. We agree with this comment. In this hospital, postoperative NRS was periodically evaluated and recorded three times a day by trained nurses. However, if the patient requires additional analysis and antiemetics, the information has been recorded in the electronic medical record and the treatment is carried out. However, as the reviewer mentioned, there may be missing time points. Considering this, we collected and analyzed the last data recorded in postoperative days (POD) 0, 1, 2, 3, 4, and 5. This is further described in lines 151-153 of this revised manuscript as follows. Please consider this point.
“NRS was investigated, which were last recorded on postoperative days (POD) 0, 1, 2, 3, 4, and 5. The requirement for rescue analgesics and rescue antiemetics was investigated during the same period.”
Point 6: Are the time points of 3 and 6 months for assessment of PSPS standard? Is it possible no differences were seen as it is too early to assess. The incidence in the cohort. Would be helpful to add to discussion.
Response 6: Thank you for pointing this out. We agree with this comment. Actually, postoperative 3 and 6 months can be said to be a short period of time to evaluate PSPS. This can also be said to be a limitation on the evaluation of PSPS. However, as described in the introduction, the authors believe that the time points set in this study are appropriate because the previous published studies also reported the results of PSPS occurrence in postoperative 3 and 6 months. Please consider this point.
Point 7: The discussion includes a lot of details that would be better suited in the introduction. Page 10 lines 236-250 should be in the introduction. The authors should use the discussion to further the implications of their results and discuss further as opposed to reporting background information.
Response 7: Thank you for pointing this out. We agree with this comment. As you commented, it might be better to describe what you pointed out in the introduction part. However, the authors believe that the introduction part has sufficient description of the background and purpose of this study. In particular, the authors believe that it has been presented to appeal to readers about the need for further effectiveness assessment of the application of CWA for pain control after thoracic surgery. In addition, the authors believe that providing more information in the introduction part, including what you pointed out, will backfire on stimulating readers' interest in the results of this manuscript. Furthermore, due to insufficient references related to this study on the effectiveness of CWA after chest surgery, we believe that the concordance or inconsistency between this study and the previous study may not be sufficiently explained in discussion if some of the discussions are moved to the introduction according to the reviewer's opinion. So, despite the reviewer's comment, the authors think it is better to keep the contents of the introduction and discussion part that have already been described without modification. Please consider this point.
Point 8: I am curious why CWA along was not included, were there too few? This would be an interesting alternate management to discuss more in the discussion. Additional sections in the discussion could be added based on the previous comments.
Response 8: Thank you for pointing this out. We agree with this comment. As you commented, the reason why patients who used CWA alone were not included in this retrospective analysis was that there were no patients. It was further stated that this was already described in the second limitation of this study, and that it was because there were no patients. This is described in lines 378-338 of this revised manuscript as follows. Please consider this point.
“Second, patients who received CWA alone were not enrolled because there were no patients who received CWA alone. Therefore,”
We look forward to hearing from you in due time regarding our submission and to respond to any further questions and comments you may have.
Sincerely,
[Sang Hun Kim, 27th, May, 2022]

Round 2
Reviewer 1 Report
Dear Authors,
I recommend this paper for publication in IJERPH.